# Learning the Unseen: Peer-to-Peer Fine-tuning of Vision Transformers

## Abstract

In this paper, we propose a distributed training framework for fine-tuning vision transformers. We address the training process in scenarios where heterogeneous data is geographically distributed across a network of nodes communicating over a peer-to-peer network topology. These nodes have the capability to exchange information with neighboring nodes but do not share their personal training data in order to maintain data privacy. Typically, training the entire vision transformer model is impractical due to computational constraints. Therefore, it is highly preferable to use a pre-trained transformer and fine-tune it for specific downstream tasks as required. In this paper, we propose a privacy-aware distributed fine-tuning method for vision transformer based downstream tasks. We demonstrate that our approach enables distributed models to achieve similar performance results as achieved on a single computational device with access to the entire training dataset. We present numerical experiments for distributed fine-tuning of `ViT`, `DeiT`, and `Swin`-transformer models on various datasets.

## 1 Introduction

Machine learning techniques have gained considerable attention recently due to their effectiveness in solving many emerging problems in computer vision, robotics, and natural language processing O'Shea & Nash (2015); Krizhevsky et al. (2012); He et al. (2016); Vaswani et al. (2017); Chung et al. (2015); LeCun et al. (2015). This increase in performance and effectiveness can be attributed to three main factors: (a) a deeper understanding of the learning methods, which has led to the development of deep learning models, (b) availability of advanced computational devices, and (c) easier accessibility to large training datasets. In particular, for the tasks related to image understanding, vision transformers have been widely successful in recent years Dosovitskiy et al. (2021); Touvron et al. (2021); Liu et al. (2021; 2022); Han et al. (2023). Training such models however is exceptionally resource-intensive since these models are trained on huge datasets containing millions of training examples and involve millions or even billions of tuning parameters. Training all these parameters requires substantial computational cost, resulting in extended training times required to achieve optimal performance. To avoid these challenges, training a model from scratch is often not ideal and already pre-trained models after some fine-tuning have been shown to work very well in many applications.

Another important consideration in many practical applications is data acquisition, i.e., heterogeneous data is acquired by geographically distributed sensors and is often not practical to bring to a single server. If these nodes are trained on local datasets, they often struggle to generalize well. For example, if one node has a dataset with images of cats and dogs, while another node possesses images of cars and ships, they may not perform well when attempting to classify all four classes combined. These local datasets however cannot be included during pre-training because (i) they often contain private information, and (ii) new data is continuously collected, making it unavailable at the time of pre-training. Since training an entire model from scratch is practically infeasible at distributed nodes, leveraging an existing pre-trained models and fine-tuning for new downstream tasks becomes very significant.

In this paper, we propose a distributed training framework for fine-tuning vision transformers in scenarios where heterogeneous data is distributed across a network of nodes. The nodes are allowed

to communicate with their neighbors but due to privacy constraints, they are restricted from sharing any personal data samples. The main contributions of this paper are listed below:

- We propose a privacy-aware distributed fine-tuning method called `P2P-FT`, which builds upon weight-mixing and gradient-sharing. Each node only shares a subset of its model parameters (which require fine-tuning), along with its local gradients (with respect to the fine-tuning parameters), with its neighboring nodes. The proposed fusion strategies enable the estimation of the global gradient and the computation of updated model parameters;

- The proposed framework can be generalized to any vision transformer with a similar structure. We provide numerical experiments on **ViT**, **DeiT**, and **Swin** transformer models. The numerical experiments highlight the performance of `P2P-FT` and compare the results with local fine-tuning methods;

- We analyze attention maps generated by `P2P-FT` and compare them with the maps generated by locally fine-tuned models and models fine-tuned on a single server with access to all data. The proposed method encourages each node to learn feature representations for unseen images and performs accurate classifications;

- The proposed method performs effectively in heterogeneous data settings and eliminates the bias caused by the non-uniform data distributions present across different computational nodes.

## 1.1 RELATED WORK

Transformers were initially introduced for applications in the field of natural language processing for machine translation Vaswani et al. (2017). Since then, they have dominated all fields of machine learning encompassing both language and vision-related tasks Lin et al. (2022); Khan et al. (2022). In contrast, earlier approaches in natural language processing predominantly relied on Recurrent Neural Networks (RNNs) He et al. (2016) and Long Short-Term Memory (LSTM) networks Hochreiter & Schmidhuber (1997); Sherstinsky (2020). These architectures were primarily used for sequence-to-sequence tasks such as machine translation, text summarization, and speech synthesis. Although useful, these models encountered challenges in capturing long-range dependencies and were susceptible to vanishing gradient problems. Transformers effectively addressed these issues using an "*attention mechanism*". This inherent self-attention capability enables them to concurrently consider the relationships and dependencies among all elements within a sequence. Consequently, transformers can comprehend how each element in a sequence can be influenced by every other element, enabling them to learn context-aware representations.

Let's consider a simple example: "*The cake came out of the oven and it tasted great.*" In this sentence, the word "*it*" can have a relationship with "*oven*" or "*cake*". We can understand that "*it*" is related to "*cake*" in the context of the sentence but due to the sequential nature and limited context window, RNNs and LSTM networks may struggle to correctly identify this relationship. However, transformers can accurately understand such contextual relation with the help of the attention mechanism. This contributes to the success of Large Language Models (LLM) that are trained on huge corpora of data. Prominent LLM architectures such as Bidirectional Encoder Representations from Transformers (BERT) and Generative Pre-trained Transformers (GPT) are fundamentally built upon the foundation of transformers models Devlin et al. (2019); Radford & Narasimhan (2018).

Although transformers achieved remarkable success in natural language processing, they encountered certain challenges when applied to image understanding problems. Transformers were originally designed to process sequences of words or tokens, where each word is often related to the neighboring words, making the attention mechanism highly effective. On the contrary, in vision tasks, data is structured as grids of pixels with no sequential order. Therefore, vision models must possess the capability to comprehend the spatial representation of different elements within the images. Earlier work on vision understanding was primarily dominated by Convolutional Neural Networks (CNNs) O'Shea & Nash (2015). The initial successful application of transformers for image classification **ViT** was documented in Dosovitskiy et al. (2021). The authors introduced an approach to divide the images into patches and used them as input for the encoder block of the transformers. Notably, these models outperformed the previous performance results achieved by CNNs, marking a significant breakthrough in the field of image classification.

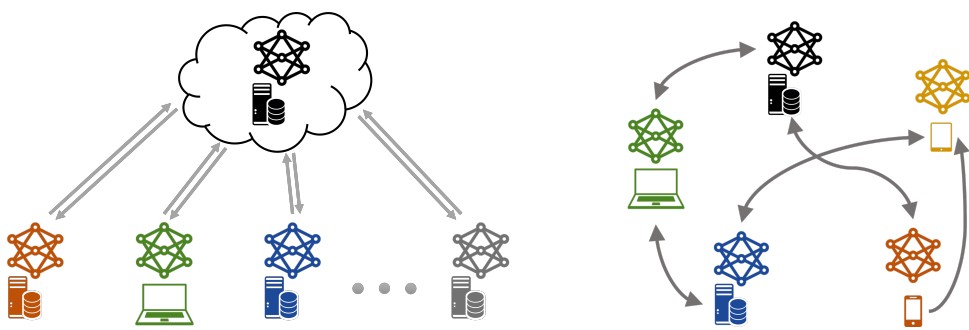

Figure 1: Distributed network topology: (Left) Hierarchical network; (right) Peer-to-peer network.

Over the past few years, vision transformers have been explored for a wide range of image understanding problems Han et al. (2023); Khan et al. (2022). However, due to the complexity of **ViT** model Dosovitskiy et al. (2021), the training process requires significant computational resources and consumes a considerable amount of time. This not only incurs substantial expenses but further has a detrimental effect on the environment. Therefore, there has been a huge emphasis on reducing the requirement of training **ViT** on large datasets. One successful attempt Touvron et al. (2021) proposed a Data-efficient image Transformer (**DeiT**) model. This model is similar to **ViT** but incorporates an additional distillation token. **DeiT** has proven to achieve competitive performance with smaller datasets, which is advantageous in scenarios where labeled data is limited. Similarly, **Swin**-Transformers, proposed in Liu et al. (2021), implement a hierarchical structure for image processing in multiple stages. They use shifted windows to efficiently understand images at various scales. While their data efficiency is comparable to **DeiT** (demanding less training data), this architecture is specifically designed for effective scaling and capturing both local and global context, particularly when dealing with high-resolution images.

Each of the above-mentioned models has gained popularity due to their promising performance. However, even the baseline models require training of more than 80 million parameters. It is generally impractical to train such models in entirety for numerous downstream tasks. Particularly for mobile devices, although there is a demand for such models, training them poses a significant challenge. To address this problem, some quantization techniques are being explored to reduce the size of vision transformers Sung et al. (2023); Frumkin et al. (2023). An alternative approach involves the utilization of pre-trained models and then fine-tuning them on specific downstream tasks using customized datasets for those particular tasks. This process leverages the representation learning capabilities of pre-trained models, simplifying the adaptation to new tasks for users. For efficient fine-tuning, the output layer is modified to match the number of classes specific to the underlying task, and it is trained while keeping most of the model's parameters frozen. Consequently, we only train a small subset of the model parameters. This training method has gained significant interest due to its ease of use and robustness Wei et al. (2022).

Vision transformers work well in addressing image understanding problems, but they encounter challenges when the data is geographically distributed over a network of computational nodes (such as cellphones, laptops, and tablets). For distributed training scenerios, different variants of two architectures are primarily explored Kairouz et al. (2021); Xin et al. (2020), i.e., (a) Hierarchical (server-client) network (see Figure 1, left), which is mainly used for federated learning problems McMahan et al. (2016); Wang et al. (2020); Karimireddy et al. (2020); Li et al. (2020); and (b) Peer-to-peer network (see Figure 1, right), which is often used in distributed optimization theory Assran et al. (2019); Ram et al. (2010); Xin et al. (2021); Qureshi et al. (2021). Hierarchical networks gained significance due to their promising results in scenarios where the network is well-structured. However, this architecture is vulnerable because any disruption to the server can compromise the entire system. To mitigate this risk, federated architectures often incorporate redundancy of servers. In contrast, peer-to-peer networks lack a single point of failure, as there is no central server node. Instead, all nodes (generally) contribute equally to the optimization task.

## 2 Preliminaries and Proposed Framework

In this section, we describe the proposed framework P2P-FT. To formulate the problem, we begin by describing the fine-tuning problem for vision transformers and then proceed to explain various distributed peer-to-peer training methodologies.

### 2.1 Fine-tuning Transformers

The transformers, initially proposed in Vaswani et al. (2017), consist of an *encoder* and a *decoder*. The encoder helps understand the representation of different features possessed by the input, while the decoder generates the sequential outputs commonly used in tasks like language translation. However, in computer vision, tasks often focus on image understanding (for example, image classification or object detection). Therefore, vision transformers utilize the encoder component of the standard transformer architecture.

Vision transformers take images as input and divide them into small patches. These patches undergo flattening through a linear projection layer, and position embeddings are added to them. Additionally, a learnable classification token is introduced into the input sequence. This modified input is then processed by the standard Transformer's encoder. Within the Transformer architecture, positional information and input features are used to apply the "*attention mechanism*", enabling the model to understand the underlying task (classification or object detection). To ensure that the output layer's size matches the number of classes, a Multi-Layer Perceptron (MLP) head is connected to the encoder's output. Transformer models typically consist of millions or even billions of learning parameters and require training on huge datasets containing millions of training instances. This extensive training process ensures robustness and diversity, resulting in promising performance results on test datasets. However, training all these parameters can be infeasible due to computational and time constraints. Recently, there has been significant interest in leveraging pre-trained models available on open-source platforms and fine-tuning them for specific tasks.

Now, we formally describe the training process. We denote the model parameters $\boldsymbol{\theta} \in \mathbb{R}^p$ and the pre-training dataset $\mathcal{D}_{pre}$. We can mathematically represent the pre-training problem as follows:

$$\min_{\boldsymbol{\theta}} \left\{ \mathbb{E}_{\mathbf{x} \sim \mathcal{D}_{pre}} L(\mathbf{x}; \boldsymbol{\theta}) \right\},$$

where $L(\mathbf{x}; \boldsymbol{\theta})$ is the function that evaluates the loss based on the training data $\mathbf{x}$ sampled from the pre-training dataset $\mathcal{D}_{pre}$. The objective is to minimize the loss by learning the optimal model parameters $\boldsymbol{\theta}^*$. After successfully learning the optimal model parameters, we proceed to fine-tune the vision transformers for downstream tasks. This involves taking the pre-trained model and modifying it by replacing the MLP head to match the number of outputs required for our specific task. Subsequently, the model undergoes training using the task-specific dataset. In the fine-tuning process, a subset of pre-trained parameters $\boldsymbol{\theta}' \subset \boldsymbol{\theta}$ are typically kept constant, while training the remaining parameters $\widehat{\boldsymbol{\theta}}$ to minimize the modified loss $\widehat{L}$:

$$\min_{\widehat{\boldsymbol{\theta}}} \left\{ \mathbb{E}_{\mathbf{x} \sim \mathcal{D}_{tune}} \widehat{L}(\mathbf{x}; \widehat{\boldsymbol{\theta}}, \boldsymbol{\theta}') \right\},$$

where the training dataset $\mathcal{D}_{tune}$ is specific to the new task. As an example, we consider the use of a pre-trained "**timm/vit_small_patch16_224**" model for classifying the CIFAR-10 dataset. We can download this model using **timm** library. However, it's important to note that the model is pre-trained on the Imagenet dataset. Therefore, to adapt it for CIFAR-10, we replace the MLP head with one of size 10, matching the number of classes in the CIFAR-10 dataset. The transformer is then trained on the downstream task using the CIFAR-10 dataset. This greatly speeds up the training process since the pre-trained model already possesses valuable knowledge of essential features from the Imagenet dataset, which are used to better understand the examples in the CIFAR-10 dataset.

### 2.2 Distributed Learning Framework

The distributed learning methods consider the problem to be divided over a network of $n$ nodes. Each node possesses its local loss function $L_i$ and some private data $\mathcal{D}_i$. The global problem is to

minimize the average of all local cost functions, i.e., for $\mathcal{D} = \{\mathcal{D}_1, \cdots, \mathcal{D}_n\}$,

$$\min_{\boldsymbol{\theta}} \left\{ L(\boldsymbol{\theta}, \mathcal{D}) := \frac{1}{n} \sum_{i=1}^{n} L_i(\boldsymbol{\theta}, \mathcal{D}_i) \right\}.$$

An intuitive solution considers the federated learning framework, where the gradients are evaluated at client nodes and then sent to the server node. The server aggregates these gradients and uses them to update the server model parameters. These updated parameters are then transmitted back to the client nodes, allowing them to update their local models using these parameters. The limitation of this approach is dependent on the reliability of the server and the constraints imposed by network connectivity. In this setup, all clients are required to maintain a bi-directional connection with the server, which can result in significant communication bandwidth demands. Additionally, the server represents a single point of failure within this architecture. In case of a server outage or an attack, all clients are adversely impacted. Hence, there is a demand for methods designed to operate within a fully distributed peer-to-peer network topology.

A well-known approach in the literature on distributed optimization considers a first-order gradient descent method `DGD` Kar et al. (2012) to minimize the loss when dealing with the data distributed over a peer-to-peer network of nodes. The nodes are prohibited from sharing private data $\mathcal{D}_i$ but can exchange their local model parameters $\boldsymbol{\theta} \in \mathbb{R}^q$ with neighboring nodes. We define $W = \{w_{i,j}\} \in \mathbb{R}^{n \times n}$ as the weight matrix representing network connectivity, where $W$ is assumed to be doubly stochastic for a strongly connected, weight-balanced graph. For each node $i$, we consider $\boldsymbol{\theta}_i^k \in \mathbb{R}^q$ as the local parameter estimate vector computed at $k$-th iteration. For simplicity of notation, we consider $L_i(\boldsymbol{\theta}_i^k) := L_i(\boldsymbol{\theta}_i^k, \mathcal{D}_i)$. Then at each iteration `DGD` computes the following:

$$\boldsymbol{\theta}_i^{k+1} \leftarrow \sum_{j=1}^{n} w_{i,j} \left( \boldsymbol{\theta}_i^k - \alpha \nabla L_i(\boldsymbol{\theta}_i^k) \right), \qquad \forall k > 0, \tag{1}$$

where $\alpha$ is the learning rate. We note that $\boldsymbol{\theta}_i^k - \alpha \nabla L_i(\boldsymbol{\theta}_i^k)$ represents a local gradient descent update, while the aggregation helps in attaining consensus over all nodes. In summary, each node tries to learn its local solution while being influenced by the parameters possessed by its neighboring nodes. This approach performs well when data distributions are homogeneous across all nodes.

However, in the majority of practical applications, data heterogeneity often leads to a notable difference between the global and the local solution. This discrepancy results in a finite gap between local and global losses, i.e., for all $i$,

$$\|\nabla L_i(\boldsymbol{\theta}) - \nabla L(\boldsymbol{\theta})\| \neq 0,$$

and this leads to inexact convergence. Figure 2 shows an example of a simple regression problem where the local solutions are very different from the global solution. To address this issue, a gradient-tracking methodology called `GT-DGD` was introduced Qu & Li (2017). This approach involves computing an additional term to estimate the gradient of the global loss function. This additional term $\boldsymbol{\tau}_i^k$ can be evaluated as:

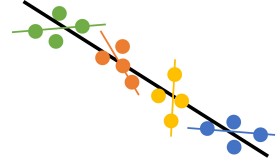

Figure 2: Different local and global solutions for a distributed regression problem.

$$\boldsymbol{\tau}_i^{k+1} \leftarrow \sum_{j=1}^{n} w_{i,j} \left( \boldsymbol{\tau}_i^k + \nabla L_i(\boldsymbol{\theta}_i^{k+1}) - \nabla L_i(\boldsymbol{\theta}_i^k) \right), \qquad k \geq 0,$$

where $\boldsymbol{\theta}_i^0 \in \mathbb{R}^p$ and $\boldsymbol{\tau}_i^0 = \nabla L_i(\boldsymbol{\theta}_i^0)$. `GT-DGD` replaces the local gradient $\nabla L_i(\boldsymbol{\theta}_i^k)$, as evaluated in equation 1, with this gradient-tracking term $\boldsymbol{\tau}_i^k$. It can be verified that at each node, $\boldsymbol{\tau}_i \rightarrow \nabla L$. Consequently, this strategy eliminates the gap between local and global losses caused by the heterogeneous data distribution. Furthermore, the distributed processing enables the network to share the computational demand, leading to faster convergence.

Another limitation of both `DGD` and `GT-DGD` is their deterministic nature, which requires the evaluation of full-batch gradients at every iteration. In many machine learning applications, dealing with huge datasets makes computing the full-batch gradients infeasible. Particularly in streaming scenarios, where data is acquired in real-time, models cannot access the entire dataset to compute the gradient. These situations demand the use of stochastic mini-batch distributed gradient descent

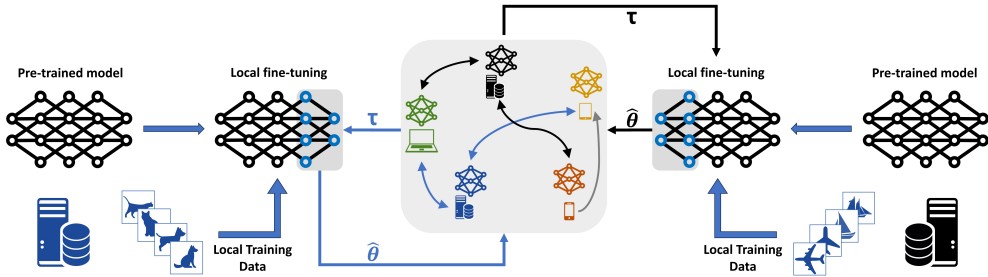

Figure 3: Proposed framework for distributed fine-tuning of vision transformer models.

methods. Furthermore, it is noteworthy that `DGD` and `GT-DGD` are primarily designed for bidirectional undirected networks and cannot be directly applied to directed networks Nedić & Olshevsky (2016); Assran et al. (2019); Xin et al. (2020). In the following section, we present our proposed framework, which utilizes distributed gradient descent with gradient-tracking for fine-tuning vision transformer models.

## 2.3 PROPOSED FRAMEWORK

Now, we describe a distributed learning method for fine-tuning transformer model parameters. Our method uses an efficient weight-mixing and gradient-sharing strategy, enabling us to achieve good performance comparable to the centralized fine-tuning setup. We define $\boldsymbol{\theta}$ as the set of all pre-trained model parameters out of which $\boldsymbol{\theta}'$ are kept constant and $\widehat{\boldsymbol{\theta}}$ are to be fine-tuned. Each node possesses a local dataset $\mathcal{D}_i \in \mathcal{D} := \{\mathcal{D}_1, \cdots, \mathcal{D}_n\}$ and a local loss function $L_i(\widehat{\boldsymbol{\theta}}, \boldsymbol{\theta}', \mathcal{D}_i)$. We note that $\boldsymbol{\theta}'$ remains fixed throughout the training process. To simplify the notation, we define $L_i(\widehat{\boldsymbol{\theta}}) := L_i(\widehat{\boldsymbol{\theta}}, \boldsymbol{\theta}', \mathcal{D}_i)$ and the global loss $L(\widehat{\boldsymbol{\theta}}) := \frac{1}{n} \sum_{i=1}^n L_i(\widehat{\boldsymbol{\theta}})$. The goal is to minimize the global loss, which can be expressed as follows:

$$\mathbf{P}: \qquad \min_{\widehat{\boldsymbol{\theta}}} \left\{ L(\widehat{\boldsymbol{\theta}}) := \frac{1}{n} \sum_{i=1}^n L_i(\widehat{\boldsymbol{\theta}}) \right\}.$$

To this aim, we propose a distributed stochastic gradient descent method that leverages weight-mixing and gradient-sharing strategies to update the fine-tuning parameters using the estimate of the global gradient. We consider $n$ nodes communicating over a strongly connected network, with $W \in \mathbb{R}^{n \times n}$ representing the connectivity matrix. In general, this matrix may not be doubly stochastic. To address the asymmetry, we define two mixing matrices $A = \{a_{i,j}\}$ and $B = \{b_{i,j}\}$, where $a_{i,j} = w_{i,j} / \sum_{j=1}^n w_{i,j}$ and $b_{i,j} = w_{i,j} / \sum_{i=1}^n w_{i,j}$ for all $i, j = \{1, 2, \cdots n\}$. This normalization ensures that matrix $A$ is row stochastic, while matrix $B$ is column stochastic. Furthermore, for each node $i$, we define $\widehat{\boldsymbol{\theta}}_i^k \in \mathbb{R}^q$ as the estimate of fine-tuning parameters evaluated at the $k$-th iteration. Figure 3 illustrates the evolution of local models using `P2P-FT` method. Depending on the type of devices, each node may differ in computational capabilities and have distinct local data classes and dataset sizes. In each iteration, every node shares its local fine-tuning parameters with its neighbors and obtains their local gradient-tracking terms. Subsequently, each node aggregates these gradient estimates and computes a weighted sum to update the local models. Algorithm 1 formally describes the `P2P-FT` method. The estimate of the fine-tuning parameters $\widehat{\boldsymbol{\theta}}_i^k$ is initialized (partially) randomly. The updates can be divided into two main steps: (i) computing the gradient-tracking term $\boldsymbol{\tau}_i^k$ to estimate the global gradient direction (line 5); and (ii) updating the local model parameters $\widehat{\boldsymbol{\theta}}_i^k$ by performing a gradient descent step and aggregating model parameters from neighboring nodes (line 2). We note that the model parameters for the MLP head are always initialized randomly, while the other parameters belonging to the layers to be fine-tuned can be initialized with the same values as those obtained from the pre-trained model. This typically leads to faster convergence as the model leverages the knowledge acquired by the pre-trained parameters.

---

**Algorithm 1** `P2P-FT` at each node $i$

---

**Require**: $\widehat{\boldsymbol{\theta}}_i^0 \in \mathbb{R}^q, \alpha > 0, \mathcal{D}_i, \{a_{ij}\}, \{b_{ij}\}$

Sample a mini-batch from $\mathcal{D}_i$ and evaluate the gradient $\boldsymbol{\tau}_i^0 := \nabla L_i(\widehat{\boldsymbol{\theta}}_i^0)$

1: **for** $k = 0, 1, 2, \cdots$ **do**
2:     $\widehat{\boldsymbol{\theta}}_i^{k+1} \leftarrow \sum_{j=1}^n a_{i \cdot j} \left( \widehat{\boldsymbol{\theta}}_j^k - \alpha \boldsymbol{\tau}_j^k \right)$
3:     Sample a mini-batch from $\mathcal{D}_i$ and evaluate the gradient $\nabla L_i(\widehat{\boldsymbol{\theta}}_i^{k+1})$
4:     $\mathbf{g}_i^{k+1} \leftarrow \nabla L_i(\widehat{\boldsymbol{\theta}}_i^{k+1})$
5:     $\boldsymbol{\tau}_i^{k+1} \leftarrow \sum_{j=1}^n b_{ij} \left( \boldsymbol{\tau}_j^k + \mathbf{g}_j^{k+1} - \mathbf{g}_j^k \right)$
6: **end for**=0

---

## 3   NUMERICAL EXPERIMENTS

In this section, we consider the distributed fine-tuning of three vision transformer architectures: **ViT**, **DeiT**, and **Swin** Transformer Dosovitskiy et al. (2021); Touvron et al. (2021); Liu et al. (2021). Each of these models consists of several attention blocks. Our approach involves freezing the model parameters for the majority of these blocks while training only a limited number of selected layers. We evaluate the classification accuracy for `P2P-FT` and compare the results with locally trained models (`Local-FT`) across multiple datasets. We perform all experiments using a HPC cluster, and for a fair comparison, we set the learning rate to $10^{-3}$.

### 3.1   NETWORK

We now describe the distributed training setup. We consider a peer-to-peer network of $n = 4$ nodes communicating over a strongly connected graph (see Figure 1 for a generic peer-to-peer network). Each node possesses its own pre-tuned model and local training dataset. These nodes can exchange fine-tuning model parameters and gradients computed during backpropagation but are prohibited from sharing their private datasets. Next, we describe the datasets on which we fine-tune our models.

### 3.2   DATASETS

We use four datasets for fine-tuning the vision transformer models and for evaluating their classification accuracy: (i) Oxford-Pets Parkhi et al. (2012), (ii) Oxford-Flowers Nilsback & Zisserman (2008), (iii) CIFAR-10 Krizhevsky et al. (b), and (iv) CIFAR-100 Krizhevsky et al. (a) datasets. Each dataset consists of colored images categorized into multiple classes. The Oxford-Pets dataset consists of 37 categories of dogs and cats with roughly 200 images for each class. The Oxford-Flowers dataset comprises images belonging to 102 flower categories, with each class containing between 40 to 258 images. The CIFAR-10 dataset contains 60,000 images belonging to 10 different categories. The CIFAR-100 dataset consists of 100 classes, each containing 600 images. Each dataset is divided into training and testing sets. The models are fine-tuned on the training set, and their accuracy is evaluated using the testing set.

| Datasets | Node 1 | Node 2 | Node 3 | Node 4 |
|:--------:|:------:|:------:|:------:|:------:|
| **Pets** | 0-4 | 5-19 | 20-29 | 30-36 |
| **Flowers** | 0-19 | 20-39 | 40-79 | 80-101 |
| **CIFAR-10** | 0-1 | 2-3 | 4-6 | 7-10 |
| **CIFAR-100** | 0-9 | 10-19 | 20-74 | 75-99 |

Table 1: Distribution of classes across different nodes.

In the distributed fine-tuning setup across a peer-to-peer network, the nodes are assigned non-overlapping classes, i.e., node 1 is never fine-tuned on the classes possessed by any other node. The class distributions for each dataset across different nodes are explicitly described in Table 1.

## 3.3 Transformer Architectures and Performance Results

We now describe the architecture of `ViT`. We use **timm/vit_small_patch16_224** which utilizes a convolutional layer to generate patch embeddings. These embeddings are then passed through a sequence of 12 blocks. Each block includes a combination of normalization, attention, and MLP layers. Finally, the output is directed to the head, which we have modified to match the number of classes in the global dataset. In our experiments, we assume that each node possesses the same pre-tuned `ViT` model as described above. At each node, we keep the weights constant for all blocks except the last one. The unfrozen weights are then updated using the method outlined in Algorithm 1. Each node possesses a non-overlapping private fine-tuning dataset comprising images from distinct classes. However, the testing dataset includes data from all classes. Table 2 highlights the accuracy achieved using the test set at each node. It can be observed that fine-tuning `Local-FT` on local data results in poor accuracy. This limitation arises because the nodes primarily focus on understanding the features of their local dataset. However, `P2P-FT` uses weight-mixing and gradient-sharing strategies to achieve superior performance, similar to centralized training setups. We now show the attention maps to gain a better understanding of the differences between `Local-FT` and `P2P-FT`.

| ViT | Local-FT | | | | P2P-FT | | | |
|---|---|---|---|---|---|---|---|---|
| **Datasets** | **Node 1** | **Node 2** | **Node 3** | **Node 4** | **Node 1** | **Node 2** | **Node 3** | **Node 4** |
| **Pets** | 13.80 | 43.10 | 23.88 | 13.13 | **87.21** | **87.28** | **87.21** | **87.21** |
| **Flowers** | 13.36 | 15.65 | 44.84 | 25.98 | **99.80** | **99.80** | **99.80** | **99.80** |
| **CIFAR-10** | 20.07 | 19.88 | 29.78 | 29.94 | **97.44** | **97.45** | **97.44** | **97.44** |
| **CIFAR-100** | 9.86 | 9.81 | 49.28 | 24.07 | **87.40** | **87.40** | **87.44** | **87.40** |

Table 2: Accuracy after fine-tuning `ViT` model for 100 epochs.

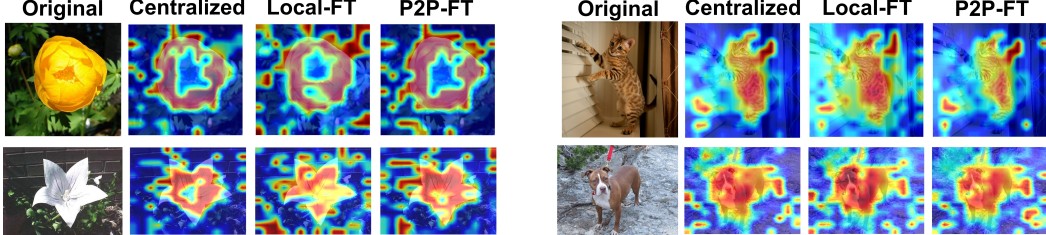

Figure 4: Visualizing attention weights for `ViT` model using Flowers and Pets datasets.

Figure 4 and 5 illustrate the attention maps overlaid on original images selected from the test set. Figure 4 presents the results when the class of each original image is present in the local fine-tuning dataset. A heat map is used to highlight the attention weights for the last block (which was fine-tuned using `Local-FT` and `P2P-FT`). The figure illustrates that the attention weights learned using `Local-FT` and `P2P-FT` are similar to what we expect from the centralized setup.

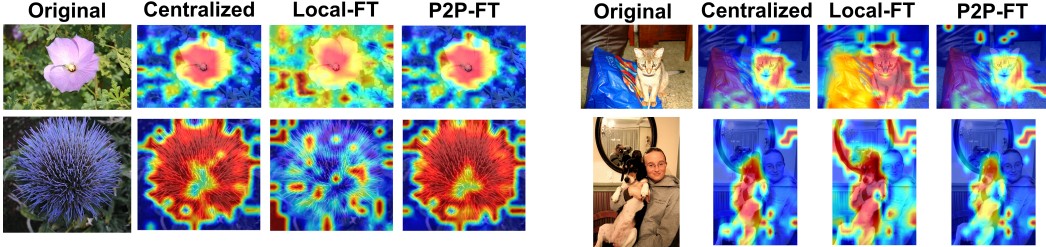

Figure 5: Visualizing attention maps for test samples belonging to unseen classes.

Figure 5 shows the attention maps learned when the class of each original image is absent from the local fine-tuning dataset. It can be observed that `Local-FT` encounters difficulties in understanding

the image and tends to apply attention towards incorrect objects. It either fails to recognize any target object or applies attention to multiple objects. However, the attention maps learned by `P2P-FT` closely resemble those learned in the centralized fine-tuning setup, despite not being trained on those classes. Although Figures 4 and 5 show results for a single node in the network, it can be verified that similar attention maps are learned by all nodes. This is because `P2P-FT` helps each node in learning feature representations for unseen images through weight-mixing and gradient-tracking strategies.

We now extend our experiments to fine-tune the distributed **DeiT** architecture. We use the pre-trained model **timm/deit_small_patch16_224** and fine-tune it for the downstream task of classifying the datasets discussed earlier in the section. Similar to **ViT**, we freeze all blocks of **DeiT** model except the last one. Table 3 presents the accuracy results after fine-tuning the **DeiT** model for one hundred epochs. It can be observed that `Local-FT` does not perform well on the test data because it is not fine-tuned on several classes belonging to the global dataset. In contrast, `P2P-FT` consistently outperforms `Local-FT` across all cases.

| **DeiT** | Local-FT | | | | P2P-FT | | | |
|---|---|---|---|---|---|---|---|---|
| **Datasets** | **Node 1** | **Node 2** | **Node 3** | **Node 4** | **Node 1** | **Node 2** | **Node 3** | **Node 4** |
| **Pets** | 13.26 | 42.96 | 24.22 | 13.06 | **86.40** | **86.40** | **86.40** | **86.40** |
| **Flowers** | 13.36 | 15.58 | 44.63 | 25.78 | **92.81** | **92.98** | **92.87** | **92.84** |
| **CIFAR-10** | 19.88 | 19.63 | 29.45 | 29.70 | **95.15** | **95.12** | **95.1** | **95.11** |
| **CIFAR-100** | 9.70 | 9.34 | 45.96 | 23.20 | **79.05** | **79.04** | **79.14** | **79.03** |

Table 3: Accuracy after fine-tuning **DeiT** model for 100 epochs.

We finally consider the **Swin** transformer architecture, which differs significantly from both **ViT** and **DeiT**. We use the **timm/swin_small_patch4_window7_224.ms_in22k** model, which is structured into four stages, each containing several blocks. We freeze the parameters corresponding to all stages except the last block of the fourth stage. Table 4 shows the accuracy results achieved after fine-tuning **Swin** transformer using `Local-FT` and `P2P-FT`. Clearly, `P2P-FT` outperforms `Local-FT` significantly because each node collaborates through the weight mixing and gradient-sharing methodologies. Furthermore, the **Swin** transformer provides better performance results as compared to the **DeiT** model, as can be expected in the centralized setup as well.

| **Swin** | Local-FT | | | | P2P-FT | | | |
|---|---|---|---|---|---|---|---|---|
| **Datasets** | **Node 1** | **Node 2** | **Node 3** | **Node 4** | **Node 1** | **Node 2** | **Node 3** | **Node 4** |
| **Pets** | 13.13 | 43.37 | 24.15 | 12.79 | **86.81** | **86.74** | **86.60** | **86.67** |
| **Flowers** | 13.36 | 15.65 | 44.90 | 25.98 | **99.83** | **99.83** | **99.83** | **99.83** |
| **CIFAR-10** | 19.98 | 19.72 | 29.72 | 29.88 | **97.59** | **97.65** | **97.55** | **97.59** |
| **CIFAR-100** | 9.86 | 9.86 | 48.79 | 24.05 | **87.04** | **86.99** | **87.03** | **87.02** |

Table 4: Accuracy after fine-tuning **Swin**-transformer model for 100 epochs.

## 4 CONCLUSION

Training large transformer models is not feasible in many applications. Fine-tuning the pre-trained models usually guarantees the best performance. In numerous practical scenarios, heterogeneous data is distributed across a network of nodes, and accumulating all the data at any central location is not possible. When the nodes fine-tune their models using only their local datasets, they struggle to generalize effectively due to the bias caused by heterogeneous data distribution. The smaller size of local datasets and the incomplete representation of all classes significantly impact the performance of vision transformers. We propose a privacy-aware distributed training framework for fine-tuning the vision transformers. The proposed method `P2P-FT` uses weight-mixing and gradient-sharing strategies to eliminate bias and achieve optimal results, even when handling unseen data from classes the node was never trained on. We illustrate the performance of `P2P-FT` for fine-tuning distributed **ViT**, **DeiT**, and **Swin** transformer models.

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
