# OpenReview forum: "Learning the Unseen: Peer-to-Peer Fine-tuning of Vision Transformers"
_ICLR.cc/2024/Conference — ICLR 2024 Conference Withdrawn Submission_

### Official Review · Reviewer_pAbZ · 2023-10-26

**Soundness:** 1 poor
**Presentation:** 1 poor
**Contribution:** 2 fair
**Rating:** 3
**Confidence:** 4

**Summary:**

This paper introduces a framework for fine-tuning vision transformers over a peer-to-peer network topology. The privacy-aware distributed fine-tuning method proposed in this work includes weight-mixing and gradient-sharing operations. It is applied to three vision transformer variants and evaluated on four downstream image classification datasets.

**Strengths:**

1. The motivation to train vision transformers distributedly is reasonable.
2. The proposed method is applicable to different vision transformers.

**Weaknesses:**

1. The proposed method seems to be just an application of the GT-DGD method to vision transformers. In the main text, it mentions newly proposed weight-mixing and gradient-sharing operations. However, I do not see the definition of such new operations. The novelty of this work is limited.
2. The presentation and organization of this paper is poor. For instance, it includes too many descriptions of vision transformers in the related work, which is not closely related to the distributed training method proposed in this work.
3. The experiments are weak. Only comparisons with local fine-tuning are shown.

**Questions:**

Could the author explain more about the difference between Decentralized Federated Learning and this work? Only Centralized Federated Learning is discussed in this work.

[1] Decentralized federated learning: Fundamentals, state of the art, frameworks, trends, and challenges. Beltrán, Enrique Tomás Martínez, et al. 2023.
[2] Decentralized Federated Learning: A Survey and Perspective. Liangqi Yuan, et al. 2023.

---

### Official Review · Reviewer_n2Zn · 2023-10-30

**Soundness:** 2 fair
**Presentation:** 2 fair
**Contribution:** 2 fair
**Rating:** 3
**Confidence:** 2

**Summary:**

This paper proposes a distributed training framework called P2P-FT for fintuning vision transformers under the setting that data is geographically distributed over a network of computational nodes and those nodes cannot share their personal training data due to privacy. The framework uses an efficient weight-mixing and gradient-sharing strategy to achieve good performance while being privacy-preserving . The authors demonstrate empirical results over 3 models (i.e. ViT, DeiT and Swin) and 4 datasets (i.e. Pets, Flowers, CIFAR-10 and CIFAR-100), which is superior than fintuning with only local data on each node.

**Strengths:**

This paper considers an important and interesting problem, i.e. distributed finetuning of vision transformers under privacy-preserving setting.

**Weaknesses:**

I believe this paper is more about distributed systems, differential privacy, and federated learning. However, I am not an expert in these areas. Therefore, I am not sure about the novelty of this paper, and I did not review the methodology part carefully. My major concerns are listed below.

1. The paper does not explain why the proposed method is specific to **fine-tuning** or **vision transformers**. Distributed optimization of deep neural networks under the privacy-preserving setting is a general problem, **neither for specific stages (e.g. pretraining or finetuning) nor architectures (e.g. transformers, RNNs, or CNNs)**. Indeed, and the proposed strategy (Sec. 2.3 on Page 6, and Algorithm 1 on Page 7) has nothing about the two aspects.

2. The paper does not clarify what the baseline setting (i.e., Local-FT) is.  Based on the context in Sec. 3.2 and Sec 3.3 ("...It can be observed that fine-tuning Local-FT on local data ..."), it uses only a local shard of the dataset on each node without inter-node communication. However, in this way, a crucial argument in the abstract (i.e. "... enables distributed models to achieve similar performance results as achieved on a single computational device **with access to the entire training dataset**.") is not justified or maybe misclaimed.

3. The paper does not compare with existing works on distributed systems, differential privacy, or federated learning. It only uses one weak baseline (i.e., Local-FT) in the experiments. Why are these works not applicable or competitive for distributed fine-tuning vision transformers?

**Questions:**

See weaknesses part.

---

### Official Review · Reviewer_WLob · 2023-11-01

**Soundness:** 2 fair
**Presentation:** 2 fair
**Contribution:** 2 fair
**Rating:** 3
**Confidence:** 4

**Summary:**

This paper proposes a peer-to-peer distributed fine-tuning framework for vision transformers. It leverages the neighboring nodes information to avoid a single point of failure in the hierarchical distributed networks.

**Strengths:**

1. The paper is easy to read
2. It also provides some experimental results on fine-tuning the ViT, DeiT, and Swin on four classification datasets including CIFAR-10 and CIFAR-100.

**Weaknesses:**

1. Peer-to-peer distributed training and gradient-tracking methodology are not new. And it's difficult to spot the novelty of the proposed approach.
2. The results in Table 2 to Table 4 are far from convincing. Besides Local-FT, hierarchical distributed training should be another baseline to be included.
3. The experiments cannot support the claim, "the proposed method performs effectively in heterogeneous data settings and eliminates the bias caused by the non-uniform data distributions present across different computational nodes" since the training and test data for each node are from the same dataset.

**Questions:**

1. How would the performances from the proposed approach compare to the hierarchical distributed training topology?
2. Can this approach generalize to the heterogenous data distribution across training nodes?

---

### Official Review · Reviewer_NQ6C · 2023-11-07

**Soundness:** 3 good
**Presentation:** 3 good
**Contribution:** 3 good
**Rating:** 6
**Confidence:** 4

**Summary:**

This paper proposes a privacy-aware distributed training framework for fine-tuning the vision transformers.

The proposed method P2P-FT uses weight-mixing and gradient-sharing strategies to eliminate bias and achieve optimal results, even when handling unseen data from classes the node was never trained on.

Then this paper illustrates the performance of P2P-FT for fine-tuning distributed ViT, DeiT, and Swin transformer models.

**Strengths:**

The proposed framework can be generalized to any vision transformer with a similar structure. We provide numerical experiments on ViT, DeiT, and Swin transformer models.

This paper analyzes attention maps generated by P2P-FT and compares them with the maps generated by locally fine-tuned models and models fine-tuned on a single server with access to all data.

The proposed method performs effectively in heterogeneous data settings and eliminates the bias caused by the non-uniform data distributions present across different computational nodes.

**Weaknesses:**

Lacking comparison to other methods with similar settings.

**Questions:**

See Weaknesses.